# Soil-transmitted helminth surveillance in Benin: A mixed-methods analysis of factors influencing non-participation in longitudinal surveillance activities

**Emma Murphy** [1‡]*, **Innocent Comlanvi Togbevi** [2‡]*, **Moudachirou Ibikounlé** [2,3], **Euripide FGA Avokpaho** [2], **Judd L. Walson** [1,4], **Arianna Rubin Means** [1,4]

1 Department of Global Health, University of Washington, Seattle, Washington, United States of America, 2 Institut de Recherche Clinique du Bénin, Abomey-Calavi, Bénin, 3 Centre de Recherche pour la lutte contre les Maladies Infectieuses Tropicales (CReMIT/TIDRC), Université d'Abomey-Calavi, Bénin, 4 The DeWorm3 Project, University of Washington, Seattle, Washington, United States of America

‡ These authors share first authorship on this work.
* emm234@uw.edu (EM); itogbevi@gmail.com (ICT)

**Data Availability Statement:** Qualitative data cannot be shared publicly because there is a chance responses may be identifiable by location.

## Abstract

### Background

Despite the significant success of deworming programs in reducing morbidity due to soil-transmitted helminth (STH) infections globally, efforts to achieve elimination of STH as a public health problem or to potentially interrupt transmission will require improving and intensifying surveillance. However, non-participation in surveillance threatens the ability of programs to adequately monitor program status and limited research has been conducted to investigate drivers of non-participation in stool-based surveillance.

### Methodology/Principal findings

This mixed-methods exploratory sequential study took place in Comé, Benin in association with the DeWorm3 Project. Six focus group discussions were conducted with individuals invited to participate in annual DeWorm3 stool surveillance. Thematic analysis was used to identify facilitators and barriers to participation and inform the quantitative analysis. A mixed-effects logistic regression model was built using baseline DeWorm3 survey data to identify factors associated with non-participation. Qualitative and quantitative findings were merged for interpretation. Among the 7,039 individuals invited to participate in baseline stool surveillance, the refusal rate was 8.1%. Qualitative themes included: community members weighing community-level benefits against individual-level risks, circulating rumors about misuse of stool samples, interpersonal communication with field agents, and cultural norms around handling adult feces. The quantitative analysis demonstrated that adults were significantly less likely to provide a stool sample than school-aged children (OR:0.69, 95%CI: 0.55–0.88), a finding that converged with the qualitative results. Individuals from areas in the highest quartile of population density were more likely to refuse to participate (OR:1.71, 95%CI:1.16–2.52). Several variables linked to community-affinity aligned with qualitative

Anonymous/redacted transcripts are available by request only. Under agreement with the IRBs of the study, data must be blinded until the study concludes. Therefore, to avoid breaching the agreement with the ethical approval bodies, data cannot be shared publicly because the study remains blinded to outcome data. Qualitative and quantitative data are available from the DeWorm3 Institutional Data Access Committee (contact via dw3data@uw.edu) for researchers who meet the criteria for access to these data.

**Funding:** JLW and ARM received the DeWorm3 study funding from The Bill and Melinda Gates Foundation (grant #OPP1129535). https://www.gatesfoundation.org/. ICT, MI, and EFGAA's research is also funded by the DeWorm3 grant as a staff members of the Benin coordinating team. The funders had no role in study design, data collection and analysis, decision to publish, or preparation of the manuscript.

**Competing interests:** The authors have declared that no competing interests exist.

results; residing mainly in the community (OR:0.36, 95%CI:0.20–0.66) and having lived in the community for more than 10 years (OR:0.82, 95%CI:0.54–1.25) decreased likelihood of refusal.

## Conclusions/Significance

Optimizing STH surveillance will require that programs reimagine STH surveillance activities to address community concerns and ensure that no subpopulations are inadvertently excluded from surveillance data.

## Author summary

Soil-transmitted helminths (STH) are a group of intestinal parasites infecting approximately 1.5 billion people globally and resulting in significant adverse health outcomes. STH surveillance is conducted across endemic regions to assess prevalence of infection, to identify areas for mass drug administration implementation, and to monitor progress. The World Health Organization targets the elimination of STH as a public health problem in endemic settings with research currently being conducted to determine the feasibility of interrupting transmission of STH. In order to optimally design and manage programs towards these goals, and to verify whether elimination of STH as a public health problem has occurred, improvements in surveillance are needed. This mixed-methods study took place in Comé, Benin in association with the DeWorm3 Project, to identify drivers of non-participation in stool-based STH surveillance. This study found that certain individuals are more likely to refuse to participate in STH surveillance activities than others, including adults, individuals in urban areas, short-term residents in communities, and those perceiving their families to not be at risk for STH. As STH surveillance is intensified, programs will need to reimagine how surveillance is conducted to address community concerns and ensure that no subpopulations are inadvertently excluded from surveillance data.

## Introduction

Globally, the World Health Organization (WHO) estimates more than 1.5 billion people are infected with soil-transmitted helminths (STH).[1,2] Individuals with moderate-to-heavy intensity infections experience adverse health outcomes including diarrhea, abdominal pain, anemia, and impaired cognitive and physical development in children [3,4]. These infections occur primarily in low- and middle-income countries across tropical and subtropical regions, and disproportionately affect low-income communities [1,2]. The WHO's current STH policy is control of STH through mass drug administration (MDA) of albendazole or mebendazole to school-age children and adults in certain high-risk groups (e.g. adolescent girls and pregnant individuals), with the ultimate target of eliminating STH as a public health problem by eliminating morbidity in these groups [1,5–7]. While targeted MDA is successful at controlling morbidity in school-aged children, community transmission continues to occur in many areas with adults serving as reservoirs of infection, particularly of hookworm infections [8–10]. Several studies are underway to determine the feasibility of interrupting transmission of STH via more intensive approaches to deworming, including community-wide MDA (cMDA), which

treats all age groups as opposed to just children and specific targeted high risk populations [11–13].

Achieving elimination of STH as a public health problem and/or interruption of disease transmission will require rethinking existing STH surveillance protocols, which are largely underspecified and left to the discretion of endemic countries to design. Community-based surveillance will be necessary for identifying individuals who may have been missed by existing programs and delineating areas with low baseline transmission where cMDA efforts may have a higher probability of interrupting transmission. New surveillance measures will also be essential for confirming elimination of STH as a public health problem or transmission interruption at the level of the implementation unit. Current diagnostics used for STH surveillance require collecting stool samples for microscopy-based assessment of infection, usually via the Kato-Katz technique, though novel approaches including qPCR are also increasingly available [14]. Both methods require collecting fecal samples.

Although broad consent and participation in surveillance activities will be necessary for gaining accurate estimates of STH prevalence, minimal research has been conducted to investigate the acceptability of stool-based surveillance and opportunities to optimize participation in surveillance activities. One study in Kenya evaluating factors influencing decisions to participate in treatment and research programs targeting STH and schistosomiasis found associations between socioeconomic status, history of disease, receipt of treatment through the program, and an understanding of the importance of the research with an individual's willingness to provide samples (urine, blood, and stool) for research purposes [15]. A Ugandan study evaluating the acceptability of sampling procedures (self-collected vaginal swabs, blood draws and stool sample collection) related to the study of human papillomavirus (HPV) vaccine efficacy in the presence of malaria and STH co-infections found that participants had positive views of providing a stool sample but were wary of vaginal swabs and blood sampling due to fears of how the samples might be misused [16]. Evidence from clinical settings in higher income countries indicate that providing stool samples is driven by perceived benefits, clear information about the process, and protocols that assuage fears around hygiene and discretion [17]. Studies of other NTD surveillance activities, including those for onchocerciasis and trachoma, indicate that less invasive sampling procedures are preferable and decisions are similarly driven by perceived benefits, and trust and understanding of the program [18,19]. However the specific factors influencing participation in STH stool-based surveillance activities are not fully understood.

High non-participation in STH surveillance activities among some communities may act as a barrier to accurate detection, diagnosis, and elimination of STH. This in turn has implications for identifying geographic areas that are ready for elimination programming, for monitoring program implementation, and for ascertaining elimination status. Understanding drivers of non-participation in STH surveillance activities is essential to identify strategies needed to improve participation in stool sampling. The purpose of this mixed-methods study is to generate evidence regarding the demographic, sociocultural, and financial factors influencing non-participation in STH surveillance activities in Comé, Benin. Findings from this study may have implications for best practices in redesigning STH surveillance should a global policy for STH transmission interruption move forward.

## Methods

### Ethics statement

Approval for the informed consent forms and research proposal was obtained from both the Human Subjects Division at the University of Washington (STUDY00000180) and Institut de

Recherche Clinique du Bénin (IRCB) through the National Ethics Committee for Health Research (002-2017/CNERS-MS and No: 031-2019/MS/DC/SGM/DFRMT/ CNERS-Ministry of Health, Benin) from the Ministry of Health in Benin prior to this study. All participants provided written informed consent after being provided an explanation of the study in local languages. All data have been de-identified. Furthermore, all researchers involved in this analysis completed human subjects research training per University of Washington requirements.

## Research Aims

This analysis used an exploratory sequential mixed methods design (QUAL ➜ QUANT) to identify factors influencing non-participation in STH surveillance activities. The qualitative analysis aimed to identify perceived facilitators and barriers to providing a stool sample. Findings from qualitative data collection were used to develop quantitative data analysis procedures (model building). The quantitative analysis aimed to explore if the facilitators and barriers identified in the qualitative data held at a population level, in addition to identifying population-level factors driving non-participation in the stool sample surveys. Qualitative and quantitative findings were merged to assess convergence and divergence of qualitative and quantitative results.

## The DeWorm3 Project

This study was nested within the DeWorm3 Project, a cluster randomized control trial based in Benin, India and Malawi that is testing the feasibility of interrupting transmission of STH.[20]. In Benin, the DeWorm3 trial is conducted in the district of Comé, which is located in the Mono Department and has a population size of 94,969 persons, per baseline study estimates. The trial site is divided into forty clusters, randomized to receive either annual MDA of albendazole to school-aged children (standard of care) or biannual cMDA of albendazole to all eligible age groups. Treatment was provided for three years (June 2018-December 2020) followed by two years of surveillance to monitor for recrudescence (2021–2022). More information about the DeWorm3 study design can be found elsewhere [11].

As part of DeWorm3 surveillance activities to monitor changes in STH prevalence and infection intensity over time, 150 individuals from each cluster were invited to participate in a longitudinal monitoring cohort (LMC). LMC participants were randomly selected at baseline from an age-stratified census (30 pre-school-age children, 30 school-age children, 90 adults) of current cluster residents [21]. LMC participants provided annual stool samples throughout the study period to assess changes in STH prevalence and rates of reinfection. Participants in control clusters (who are not routinely offered treatment via the DeWorm3 Project) who were found to have moderate-to-heavy intensity of STH infections via Kato-Katz were followed-up with and offered deworming treatment. This analysis uses data from the baseline LMC in Benin, with consenting procedures and sampling occurring March-May 2018.

## Qualitative methods

Six focus group discussion (FGDs) were conducted with individuals invited to participate in the first LMC to understand factors influencing their decision to provide a stool sample for surveillance in December 2019. Purposive sampling was used to sample individuals from DeWorm3 clusters with high refusal rates (>7.5% refusal) and low refusal rates (<4.5% refusal). Sampling lists of refusals and non-refusals were generated for the selected clusters and the study team contacted those individuals via phone and/or household visit. Three FGDs were conducted in each setting (high refusal and low refusal clusters). Both adults invited to participate in the LMC and adult parents or guardians of children invited to participate in the

**Table 1. Sampling frame.**

| Cluster Type | Focus Group Population | Number of Clusters Sampled | Number of Sampled Individuals | Sampling Frame | Number of FGD participants |
|---|---|---|---|---|---|
| Low refusal to participate in STH surveillance | Adults (male) | 16 | 20 | Invited to participate in LMC | 8 |
| | Adults (female) | | 20 | Invited to participate in LMC | 5 |
| | Parents/guardians | | 20 | Child invited to participate in LMC | 5 |
| High refusal to participate in STH surveillance | Adults (male) | 5 | 20 | Invited to participate in LMC | 6 |
| | Adults (female) | | 20 | Invited to participate in LMC | 5 |
| | Parents/guardians | | 20 | Child invited to participate in LMC | 7 |

LMC were sampled. For each FGD, quota sampling was used to ensure participants included a balance of both people who provided a stool sample during the LMC and people who refused, either at the time of recruitment or the time of sample collection. FGD respondents in high refusal clusters were sampled from five different study clusters. FGD respondents in low refusal clusters were sampled from sixteen clusters in order to identify enough people who refused to provide a stool sample in these settings.

Adult FGDs were divided by gender to create a more comfortable atmosphere for participants to discuss their personal experiences with stool sampling. FGDs with parents/guardians were not divided by gender. Twenty individuals were sampled for each FGD. Many invited participants were unwilling or unable to participate in the scheduled discussion, resulting in FGDs consisting of 5–8 participants. Of the FGD participants, 2–4 in each group refused to provide a stool sample. Table 1 presents a summary of the sample and sampling frame.

FGDs were conducted using a semi-structured question guide informed by the Theory of Planned Behavior (TPB), which has been used previously for similar research questions [17,22]. Fig 1 presents the TPB as applied to this study. The question guide (S1 Appendix) was piloted by the Benin research team outside of the selected study areas and adapted thereafter. All FGDs were led by members of the Benin research team trained in FGD facilitation and conducted in the local language (Pédah or Mina) or in French. All the FGDs were audio-recorded following written consent from participants. Audio data from the FGDs were transcribed into French and underwent quality control checks by the Benin research team to ensure cultural nuances were maintained in the French translation. The French transcripts were then translated into English by a research team member outside Benin. Afterwards, a final round of quality control checks on the French-to-English translation were conducted.

English FGD transcripts were coded using the qualitative analysis software ATLAS.ti 9. A preliminary codebook was developed in advance, informed by the TPB. A mixture of deductive and inductive coding techniques was used to allow for iterative adaptations to the codebook. See S2 Appendix for codebook. Each transcript was coded initially by a primary coder (EM) and validated by a secondary coder (ICT) who reviewed all coding decisions, added new codes, or removed codes. A standardized validation tracker was used to record instances of disagreement and a third researcher (ARM) served as a tiebreaker as necessary. The coding team met frequently to reflect on their unique positionalities in relation to the study population; the understanding of local culture provided by the coder based in Benin (ICT) was influential in defining the final themes. Thematic saturation in coding was reached when no new emerging codes or themes were identified across coders. Once all FGD transcripts were coded,

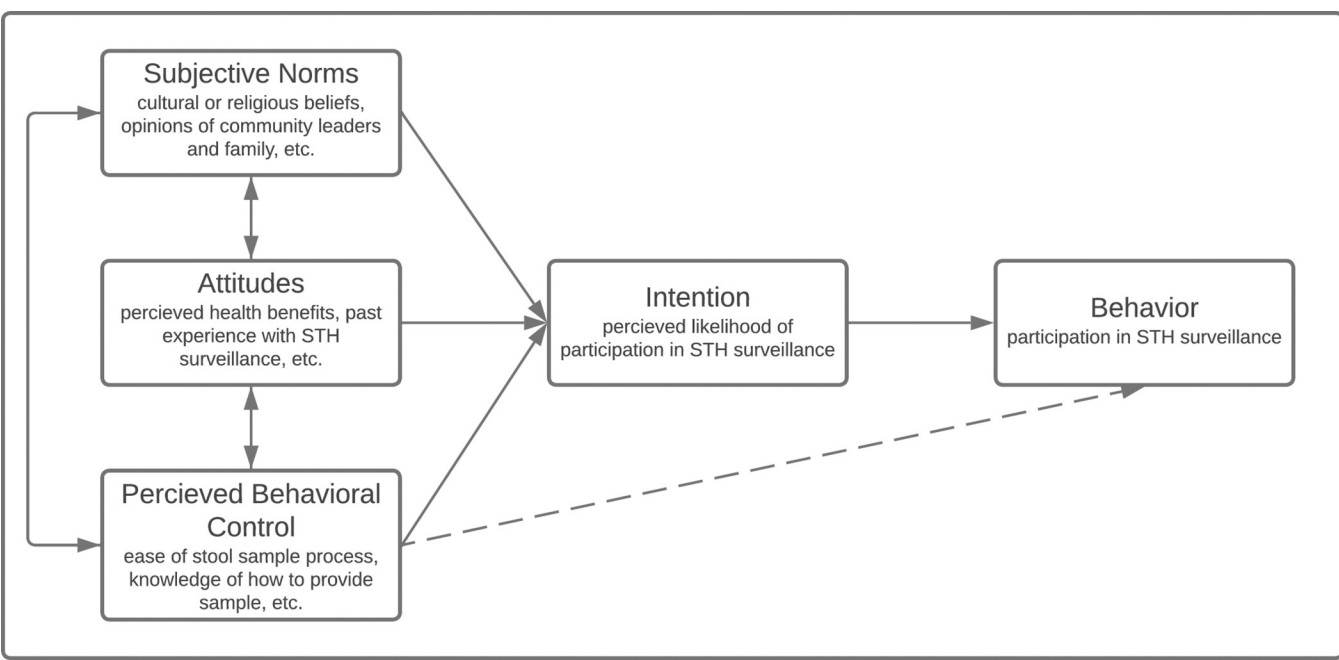

**Fig 1. Theory of planned behavior conceptual framework applied to STH surveillance participation.**

case memos summarizing emerging themes and noting exemplary quotes were developed jointly by the two coders for each setting (high and low refusal clusters) to identify salient themes.

## Quantitative methods

This analysis included 7039 individuals invited to participate in the baseline LMC survey in Benin. Twenty-two observations (0.31%) were excluded from the analysis due to incomplete data for the outcome of interest (n = 16) or the population density variable (n = 6). These 7039 individuals represented 5821 households as multiple individuals could be sampled from the same household. Demographic information was ascertained during a baseline census of all households in the study catchment area and/or via a baseline survey administered during the first LMC survey. Individuals that consented to participate in the LMC were asked to provide a fecal sample at the time of the baseline survey.

We conducted an exploratory analysis to identify the prevalence and factors associated with non-participation in the baseline LMC in Benin. The primary outcome of interest was a binary participation variable. Non-participation indicates if the individual either: a) did not consent to provide a stool sample, or b) consented but did not provide a sample at the time of collection. We present descriptive statistics (proportions, means, and standard deviations) of sampled individuals who did and did not refuse to provide a sample, separately by age, sex, SES, religion, language, education, household toilet type, time in current residence, place of main residence, population density, village population size, and baseline cluster STH prevalence. The SES variable is an asset-based index compiled using principal component analysis, following the procedure described in the Demographics and Health Survey [23,24]. Either a Pearson's Chi-squared test or a Welch Two Sample t-test was used to test the associations between dependent and independent variables.

We conducted backwards stepwise model building to identify factors associated with non-participation in baseline stool sampling. The choice of independent variables considered for

inclusion in the analysis were based on available data, existing literature, and a hypothesis-driven conceptual framework driven by the qualitative thematic analysis (S1 Fig). We then developed a mixed-effects logistic regression model using all non-collinear variables of interest [25]. Collinearity of variables was assessed using a mixture of Pearson's correlation coefficients (0.8 cut-off), chi-square tests, and knowledge of data architecture, as appropriate. The full model was then simplified using backward stepwise elimination. We compared the Akaike information criterion (AIC) of the full model to the AIC of models in which each variable, separately, was dropped [26]. The model was adjusted for Cluster ID during each step. The reduced model with the lowest AIC was kept and the process was repeated until AIC was no longer further reduced in the adjusted model. Adjustments for multiple comparisons were made by controlling the false discovery rate [27]. We present results from the fully adjusted model: odds ratios, 95% confidence intervals and the adjusted p-value. All statistical analyses were conducted using R and RStudio software (version 4.1.2) [26].

## Results

Among the 7,039 individuals invited to participate in the baseline LMC in Benin, the overall refusal rate for baseline stool sampling was 8.14% (573 individuals). Of these, 231 individuals did not consent to the stool sampling and 342 refused to provide a sample at the time of collection despite previously consenting. Overall cluster-level refusal rates ranged from 0.6% to 20% (Fig 2).

### Qualitative results

During six FGDs with 36 total participants, community members reported several factors that influenced their decision to participate in stool surveillance for STH. Main themes in their responses included: weighing of community-level benefits against individual-level risks of STH surveillance, circulating rumors about misuse of stool samples, interpersonal communication with field agents, and cultural norms around handling feces, particularly among adults.

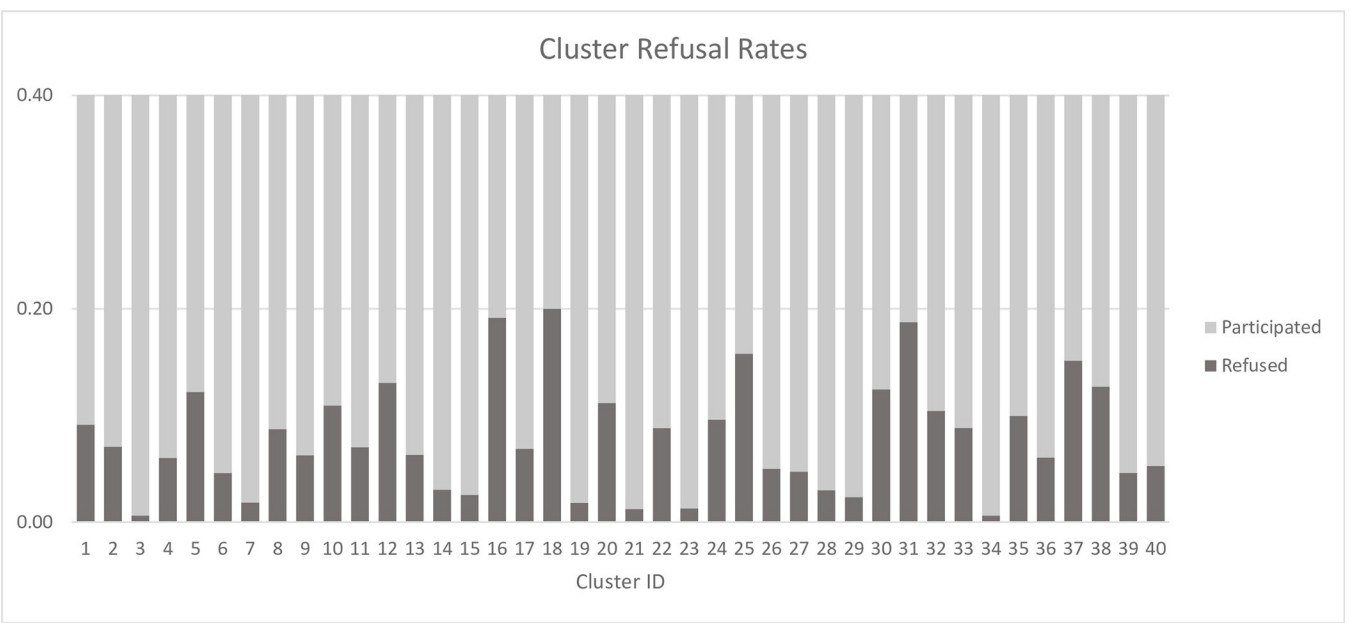

**Fig 2. Cluster-level refusal rates.**

In general, FGD respondents from clusters with low refusal rates reported fewer negative perceptions of stool surveillance.

## Across all clusters, community members weighed community-level benefits against perceptions of individual-level risks

Individuals' decisions to provide a stool sample were made based on the balance of perceived risks and benefits, including both individual-level and community-level risks and benefits. Generally, most of the risks cited by respondents were at the individual level while most of the perceived benefits were at the community level. The main perceived benefit of participating in stool surveillance was the belief that it would lead to a reduction in STH infections, which would positively impact the community collectively. There was recognition that little individual benefit would be gained from participation in surveillance activities but that the diagnostic information would help inform the program's treatment strategy. FGD participants described their willingness to provide a stool sample, despite a general aversion to handling feces, for the good of their communities. These statements were connected to an awareness that their families and community suffered from intestinal worms. Parents particularly reported the belief that participating would contribute to the reduction of STH symptoms for their children.

*"We agreed because intestinal worms destroy human health. It's also a good project brought in. It's to help the population. People care about us, that's why this project is being proposed to us. That's why we agreed." [Respondent 5, Parents FGD, Low Refusal Cluster]*

*"I told them in my turn, that it is certainly to cure a disease, because according to the explanations of the workers of the project, it is to reduce the intestinal worms that are causing enough havoc in the commune and that is why I participated by providing my stools," [Respondent 3, Women FGD, High Refusal Cluster]*

However, while recognizing the benefits to the community, FGD respondents also reported individual-level concerns about providing a stool sample. Respondents reported fear of their stool being misused and resulting in illness or death, a belief supported by circulating rumors. Such comments were voiced particularly by men in high refusal clusters. Respondents, predominantly in high refusal clusters, also expressed hesitancy due to stigma they expected to experience in their families or communities if they provided a stool sample. Women in both settings cited a cultural belief that if an adult handled their own feces it would result in swelling of the abdomen.

*"Even eating is a problem, that's why people refused to provide stool samples. Because they suspect that they will mysteriously disappear after they provide their stool samples or people will use them to do something they don't know, and many people really had this fearful idea." [Respondent 4, Women FGD, High Refusal Cluster]*

*"It can happen that we suffer from swelling if we defecate and take a sample of our own feces. That is why I refused to provide a sample." [Respondent 5, Women FGD, Low Refusal Cluster]*

While respondents from high and low refusal clusters cited similar risks and benefits, high refusal cluster FGDs discussed the individual-level risks with greater frequency. This suggests that in high refusal clusters, perceived individual-level risks influenced decisions more strongly than perceptions of community-level benefits. Additionally, in high refusal clusters several respondents expected to receive diagnostic results from their stool tests and viewed this as an

individual-level benefit. Respondents reported that not receiving the results sowed skepticism in the program and led to future non-participation.

> *"Why do adults agree to participate and provide bowel movements, but no longer do so? This is related to the fact that the record of test results is no longer being sent so that people know what they have been diagnosed with. From that point of view and after thinking about it, they no longer perform the stool donation, because they simply ask: what's the point of providing the stool if the test results are not given to them?"* [Respondent 3, Women FGD, High Refusal Cluster]

## Rumors of misuse of stool samples, compounded by a lack of familiarity with stool sample provision in household setting, were more common in high refusal clusters

When individuals did not understand the rationale for collecting stool samples, they were more likely to choose not to participate in the survey. FGD participants noted many rumors circulating in the community at the time of LMC implementation. These rumors were reported in both settings (high and low refusal clusters), however the high refusal cluster respondents commented on rumors with greater frequency than respondents in low refusal clusters, who generally described rumors as influencing others and not themselves. Similarly, statements about the "unknown purpose" of stool sampling were cited with greater frequency (approximately double) in the FGDs with high refusal cluster residents, which helped solidify belief in some of the circulating rumors. Commonly cited rumors included a belief that occult practices would be performed with stool samples, which could lead to illness and death. A few FGD participants said that the stool surveillance was affiliated with white foreigners with harmful intentions, such as lowering the birth rate in Africa. FGD participants reported hearing about these rumors from neighbors, community leaders, and social media.

> *"We see on the WhatsApp most of the serious events happening in the world. These events are broadcast via WhatsApp, and we see them. For example, you may see someone vomiting blood and dying moments later. All of this is what makes some people scared."* [Respondent 4, Women FGD, Low Refusal Cluster]

> *"They are afraid, they are afraid just because the stool comes from the human being for an unknown destination, are they going to use it for our well-being or to harm us? Really, we're all scared on that level."* [Respondent 5, Parents FGD, High Refusal Cluster]

Respondents across all FGD reported little (to no) prior experience providing stool samples, especially outside of a hospital setting; this compounded misunderstanding or skepticism about the purpose and use of stool samples. Respondents were more comfortable with the idea of collecting stool samples for the first time at the hospital as opposed to at the household level. As a result, within high refusal clusters, respondents expressed concerns that the program might be disconnected from the local health system. This disconnect gave credit to rumors of misuse.

> *"What's all this stuff? Is it the deworming that requires all this? But I was told that some diseases require stool samples to be taken for analysis to find out. . . I said I've never seen that. You see! You see! The fact that I've never seen certain things makes it difficult."* [Respondent 5, Parent FGD, Low Refusal Cluster]

*"I'm afraid that the hospital is not involved."* [Respondent 2, Parents FGD, High Refusal Cluster]

### Clear communication messages increased acceptability of providing a stool sample, with stool collectors' communication style strongly influencing community member willingness to participate

Field agents provided information to community members about the purpose of STH surveillance, how to take a stool sample, and key messages to directly address rumors. FGD participants reported that the communication styles and messages of the field agents strongly influenced community member decisions to participate in stool surveillance. Respondents noted that they were more likely to participate when the collectors were patient in explaining the benefits of the program and willing to accommodate household preferences for the timing of stool collection. However, not all FGD respondents reported positive interactions with the project agents, noting instances of perceived impoliteness or agents who were unable to answer questions about the stool sampling with confidence. These experiences negatively impacted willingness to provide a sample. Respondents in high refusal clusters had stronger and more polarized opinions of project agents and more frequently described negative experiences.

*"What I liked about these agents, they don't get angry but adopt a gentle character that makes you provide the stool sample even if you don't want to. That's what I liked about them."* [Respondent 2, Men FGD, High Refusal Cluster]

*"OK, first of all, agents who go into the field don't give a convincing speech that appeals to the people they're targeting."* [Respondent 6, Parent FGD, High Refusal Cluster]

### There is less aversion to handling children's feces, leading to increased willingness to provide stool samples for children

Almost all FGD participants reported a general aversion to handling feces due to sanitation concerns and its odor. Respondents described having to have "courage" to provide a sample, suggesting this presented a substantial hurdle that had to be overcome through understanding of the sample's purpose and benefits. However, in all FGDs, respondents stated it would be fine (even easy) to give a child's sample. They reported being less likely to provide a sample themselves, as an adult, but consenting to provide one for their young children. Female respondents directly tied this difference in willingness to handle the feces of children to the practice of cleaning up after infants and toddlers. Respondents also cited a common belief that if an adult handles their own feces they will experience swelling of the abdomen.

*"If it was even a stool sample from a child that was asked for, it would be easier to take the sample, but me at my age! No! It's not easy to take a stool sample."* [Respondent 4, Men FGD, High Refusal Cluster]

*"It makes you suffer from swelling. Or that's what our grandmothers say. . .Making your own bowel movements and picking them up yourself, it's not good. Our grandmothers used to say that, and we heard it. That's what we do so that when a child defecates, he doesn't pick it up himself. . .don't you see that because it was like that in our mentality, it's bad to pick up your own stool? If you have to collect some for your child, it's easy. But if you have to collect your own stool, it's bad. It's a bit difficult."* [Respondent 5, Parents FGD, Low Refusal Cluster]

### Participant recommendations to improve STH surveillance activities focused on communication and community engagement strategies

Most of the recommendations for improving stool surveillance across all sampled clusters involved increased communication and awareness raising strategies to explain the benefits of stool sampling and assuage people's fears. Some respondents recommended changing the stool sample kit design, such as by increasing the size of the sample receptacle and including gloves, however these recommendations were not described as essential for increasing participation. In discussing communication strategies, FGD respondents suggested using radio advertisements, town criers, community leaders, and local residents who've already provided a sample to educate community members about the benefits of the program and the need for STH surveillance. An emphasis on combatting rumors was suggested for the message content. This community engagement was highlighted as necessary for increased acceptability of and participation in STH surveillance.

> *"The second thing is to go and explain and inform via the radio so that people who had not provided their stool before can do so. . .In my opinion, those who are stubborn and have not done so before, if we can go through the radio channel, explain to them that they are invited to give their stool, that it is for diagnostic research purposes to determine something, and not for an undisclosed use."* [Respondent 1, Women FGD, High Refusal Cluster]

> *"In my opinion, if we can contact the village chiefs and town criers who will be able to pass on and re-disseminate information to the people, ringing the gong; raising their awareness; and further explaining to them how this project is important for our well-being. . .Basically, you have to be able to talk to them in such a way that they agree it very much by changing their intention towards the project."* [Respondent 5, Men FGD, High Refusal Cluster]

FGD participants in high refusal clusters also recommended working closely and more visibly with local health facilities and health workers, including community health workers, to increase the perceived legitimacy of the stool surveillance. Including local community members on the sample collection teams was emphasized as important for increasing acceptance. Respondents also suggested increasing the training for project agents, so they are better equipped to patiently address concerns of participants at the time of consent and stool collection. Lastly, FGD participants emphasized the importance of explaining whether diagnostic results will be shared after the stool is tested and providing a clear rationale for why results will not be shared. This explanation will assuage fears of misuse, reduce dissatisfaction, and increase acceptance of the surveillance activities.

> *"What I still have to say is that the villages and regions where we want to send agents should have local people in the teams. So, it won't just be foreigners; otherwise, it will be difficult, but if there are local people in the teams, that is, if there are two people, there will be one local person to join a foreigner to lead the operation. But let's make sure it's not just foreigners who don't master the life of the area. Otherwise, it will be a bit difficult."* [Respondent 5, Parents FGD, Low Refusal Cluster]

### Quantitative results

The influence of 12 variables on stool sample non-participation were considered (Table 2). Four of the variables (population density, village population size, household toilet type, and cluster-level baseline STH prevalence) were included based on qualitative findings. Population

**Table 2. Descriptive statistics.**

| Characteristic | Consented and provided stool sample N = 6,466[i] | Refused to consent or provide stool sample N = 573[i] | p-value[ii] |
|---|---|---|---|
| Sex | | | 0.384 |
| Male | 2,986 (91.5%) | 276 (8.5%) | |
| Female | 3,480 (92.1%) | 297 (7.9%) | |
| Age | | | 0.002 |
| PSAC[iii] (1–4 yrs) | 1,258 (92.7%) | 99 (7.3%) | |
| SAC[iv] (5–14 yrs) | 1,374 (93.7%) | 93 (6.3%) | |
| Adult (15+ yrs) | 3,834 (91.0%) | 381 (9.0%) | |
| Socioeconomic status (in quintiles, where 1 is lowest) | | | <0.001 |
| 1 | 1,035 (94.0%) | 66 (6.0%) | |
| 2 | 1,088 (92.8%) | 85 (7.2%) | |
| 3 | 1,251 (92.9%) | 96 (7.1%) | |
| 4 | 1,443 (91.6%) | 132 (8.4%) | |
| 5 | 1,649 (89.5%) | 194 (10.5%) | |
| Household language | | | <0.001 |
| Pédah | 1,609 (96.1%) | 65 (3.9%) | |
| Sahoué | 858 (92.1%) | 74 (7.9%) | |
| Watchi | 1,860 (90.8%) | 188 (9.2%) | |
| Mina | 1,078 (91.2%) | 104 (8.8%) | |
| Other | 1,061 (88.2%) | 142 (11.8%) | |
| Household religion | | | <0.001 |
| Christianity | 4,026 (90.8%) | 406 (9.2%) | |
| Islam | 135 (87.1%) | 20 (12.9%) | |
| Voodoo | 1,650 (95.3%) | 81 (4.7%) | |
| Other traditional religion | 125 (93.3%) | 9 (6.7%) | |
| None | 318 (90.3%) | 34 (9.7%) | |
| Other | 210 (90.1%) | 23 (9.9%) | |
| Household toilet type | | | <0.001 |
| Flush toilet | 470 (88.0%) | 64 (12.0%) | |
| Pit latrine | 3,348 (90.1%) | 369 (9.9%) | |
| No toilet facility/Bush/Field | 2,141 (95.5%) | 101 (4.5%) | |
| Other | 504 (92.8%) | 39 (7.2%) | |
| Highest education achieved by consenter | | | <0.001 |
| No education | 2,438 (92.1%) | 210 (7.9%) | |
| Primary | 1,491 (94.0%) | 96 (6.0%) | |
| Secondary | 1,232 (91.7%) | 111 (8.3%) | |
| Higher secondary | 620 (90.1%) | 68 (9.9%) | |
| College/Diploma/University | 515 (88.3%) | 68 (11.7%) | |
| Unknown | 166 (89.2%) | 20 (10.8%) | |
| Time in current residence | | | <0.001 |
| <1 year | 378 (89.8%) | 43 (10.2%) | |
| 1–5 years | 1,770 (90.0%) | 197 (10.0%) | |
| 6–10 years | 1,418 (90.4%) | 150 (9.6%) | |
| 11–20 years | 1,210 (93.7%) | 82 (6.3%) | |
| >20 years | 1,643 (94.3%) | 99 (5.7%) | |
| Don't know | 47 (95.9%) | 2 (4.1%) | |

(*Continued*)

**Table 2.** (Continued)

| Characteristic | Consented and provided stool sample N = 6,466[i] | Refused to consent or provide stool sample N = 573[i] | p-value[ii] |
|---|---|---|---|
| Live here majority of days past 6 months | | | <0.001 |
| Yes | 6,411 (92.0%) | 557 (8.0%) | |
| No | 55 (77.5%) | 16 (22.5%) | |
| Population density w/in 1km of household (in quartiles)[v] | | | <0.001 |
| 3–404.5 persons/km$^2$ | 1,653 (93.8%) | 110 (6.2%) | |
| 404.5–817 persons/km$^2$ | 1,635 (93.1%) | 122 (6.9%) | |
| 817–1,480 persons/km$^2$ | 1,603 (91.1%) | 156 (8.9%) | |
| 1,480–2,528 persons/km$^2$ | 1,575 (89.5%) | 185 (10.5%) | |
| Village population (in quartiles)[v] | | | <0.001 |
| 1,379–7,600 | 1,742 (94.6%) | 100 (5.4%) | |
| 7,600–11,000 | 1,606 (90.2%) | 175 (9.8%) | |
| 11,000–12,500 | 1,562 (89.7%) | 179 (10.3%) | |
| 12,500–18,657 | 1,556 (92.9%) | 119 (7.1%) | |
| Baseline cluster-level STH prevalence | 5.36 (7.59)[vi] | 3.70 (5.21)[vi] | <0.001[vii] |

*i n (%)*
*ii Pearson's chi-squared test*
*iii PSAC: Pre-school aged children*
*iv SAC: School-aged children*

*v Data generated from the DeWorm3 census & geospatial data*
*vi mean (SD)*
*vii Welch two sample t-test*

density and village population size were included because qualitative findings indicated that the degree of solidarity between community members might increase perceived community-level benefits of STH surveillance. Household toilet type and cluster-level baseline STH prevalence were included because qualitative data suggested that an individual's experience with STH or perceived risk of STH infection increased the likelihood of providing a stool sample.

An individual's age, socio-economic status, household religion, household language, consenter's education, household toilet type, time in current residence, place of main residence, population density, village population size, and baseline cluster STH prevalence were significantly different amongst individuals who refused to participate versus those who participated in STH surveillance (Table 2).

The full set of variables (Table 2) were included in a mixed-effects logistic regression model, with cluster ID included as a random effect. After five rounds of backwards stepwise elimination, no further reduction in AIC was reached by removing remaining variables (S3 Appendix). The final reduced model included age, household language, consenter's education, time in current residence, place of main residence, population density, and cluster-level baseline STH prevalence. Odds ratios, confidence intervals, and adjusted p-values for the reduced model are reported in Table 3. Overall, while there were statistically significant differences between non-participants and participants, the actual magnitude of difference was not large for most variables.

Results demonstrate that adults were more likely to refuse to provide a stool sample than pre-school and school-age children (OR:0.79, 95% CI:0.63–1.00 and OR:0.69, 95% CI:0.55–0.88, respectively). It is important to note that both pre-school and school-age children were unable to consent for their own participation but required parental consent (under age 7) or assent (ages 7–17) to provide a stool sample. This finding aligns with qualitative data that suggest it is more acceptable to provide a child's stool sample than an adult's. Individuals from households who speak a language other than Pédah, the language most commonly spoken in

**Table 3. Reduced model.**

| Predictors | Refusal to Provide Baseline Stool Sample | | |
|---|---|---|---|
| *Predictors* | *Odds Ratios* | *CI* | *adjusted p-value[i]* |
| **Age** | | | |
| Adult (15+yrs) | Reference | | |
| Pre-school age children (1-4yrs) | 0.79 | 0.63–1.00 | 0.111 |
| School-age children (5-14yrs) | 0.69 | 0.55–0.88 | 0.016 |
| **Household language** | | | |
| Pédah | Reference | | |
| Sahoué | 1.99 | 1.33–2.98 | 0.007 |
| Watchi | 1.68 | 1.16–2.43 | 0.021 |
| Mina | 1.44 | 0.98–2.12 | 0.126 |
| Other | 1.96 | 1.35–2.85 | 0.007 |
| **Consenter education level** | | | |
| None | Reference | | |
| Primary | 0.68 | 0.53–0.88 | 0.017 |
| Secondary | 0.90 | 0.70–1.15 | 0.508 |
| Higher secondary | 0.96 | 0.71–1.30 | 0.800 |
| College/University | 1.13 | 0.84–1.53 | 0.524 |
| Unknown | 1.12 | 0.68–1.84 | 0.698 |
| **Time in current residence** | | | |
| <1yr | Reference | | |
| 1-5yr | 1.09 | 0.75–1.56 | 0.698 |
| 6-10yr | 1.12 | 0.77–1.64 | 0.646 |
| 11-20yr | 0.82 | 0.54–1.25 | 0.497 |
| >20yr | 0.76 | 0.51–1.15 | 0.300 |
| Don't know | 0.37 | 0.08–1.62 | 0.300 |
| *Predictors* | *Odds Ratios* | *CI* | *adjusted p-value[i]* |
| **Participant lives in community majority of days during past six months** | | | |
| No | Reference | | |
| Yes | 0.36 | 0.20–0.66 | 0.007 |
| **Population density w/in 1 km of HH (in quartiles)** | | | |
| 1$^{st}$ quartile (3–404.5 persons/km$^2$) | Reference | | |

(*Continued*)

**Table 3.** (Continued)

| | | | |
|---|---|---|---|
| 2$^{nd}$ quartile (404.5–817 persons/km$^2$) | 1.40 | 1.02–1.93 | 0.102 |
| 3$^{rd}$ quartile (817–1,480 persons/km$^2$) | 1.45 | 1.01–2.08 | 0.110 |
| 4$^{th}$ quartile (1,480–2,528 persons/km$^2$) | 1.71 | 1.16–2.52 | 0.021 |
| Baseline Cluster STH Prevalence | 0.98 | 0.95–1.01 | 0.260 |

*i*: p-values adjusted to control for the false discovery rate per Benjamini & Hochberg [27,28]

rural areas of Comé, were more likely to refuse to provide a stool sample than those who speak Pédah at home (Table 3). Individuals who had lived in their place of residence for the majority of days during the 6 months preceding the LMC survey were less likely to refuse to provide a sample than others for whom their home had not been their main place of residence preceding the survey (OR:0.36, 95%CI: 0.20–0.66).

These findings indicate that if someone has resided in their community for more than ten years, they are less likely to refuse to provide a stool sample (Table 3). For example, individuals who have resided in their community for at least 20 years were less likely to refuse to provide a sample compared to those who've been living in their communities for less than a year (OR:0.76, 95%CI: 0.51–1.15). Additionally, as the population density within one kilometer of an individuals' household increased, they were more likely to refuse to provide a stool sample (Table 3). Lastly, refusals decreased with increasing baseline STH cluster prevalence (OR:0.98, 95%CI:0.95–1.01). While this finding is not statistically significant and not of high magnitude, it aligns with qualitative findings about the relationship between experience with STH symptoms and willingness to provide a stool sample.

Additionally, based on qualitative findings that suggested individuals' perceptions of and intentions to provide stools samples change over time, we examined longitudinal participation data (Fig 3). Longitudinal data demonstrated a general increase in refusals over time, with a large drop-off in participation after the initial baseline survey. Important to note, some portion of the study population who refused to provide a sample at baseline later chose to and vice versa, suggesting changes in individuals' decisions over time.

## Discussion

STH surveillance programs must achieve both high coverage and representative sampling in order to optimally measure progress. However, surveillance can be limited by both low levels of participation and systematic non-compliance. Among the DeWorm3 trial sites, baseline refusals rates among individuals asked to provide a stool sample ranged from approximately 8% (Benin and India) to 38% (Malawi). Notably, these refusals were observed in a trial setting, suggesting refusals could be even higher in routine public health surveillance programs. Despite this, there is limited evidence in the published literature regarding the acceptability of stool-based surveillance, and factors that influence participation. Motivated by observed refusals to consent to STH surveillance activities associated with the DeWorm3 trial in Comé, Benin, this study aimed to understand determinants of these refusals. This paper presents a mixed-methods analysis of factors influencing non-participation in stool sample-based STH surveillance programs.

This study identified several individual-level characteristics that influenced non-participation in STH surveillance surveys. Both qualitative and quantitative data demonstrated adults

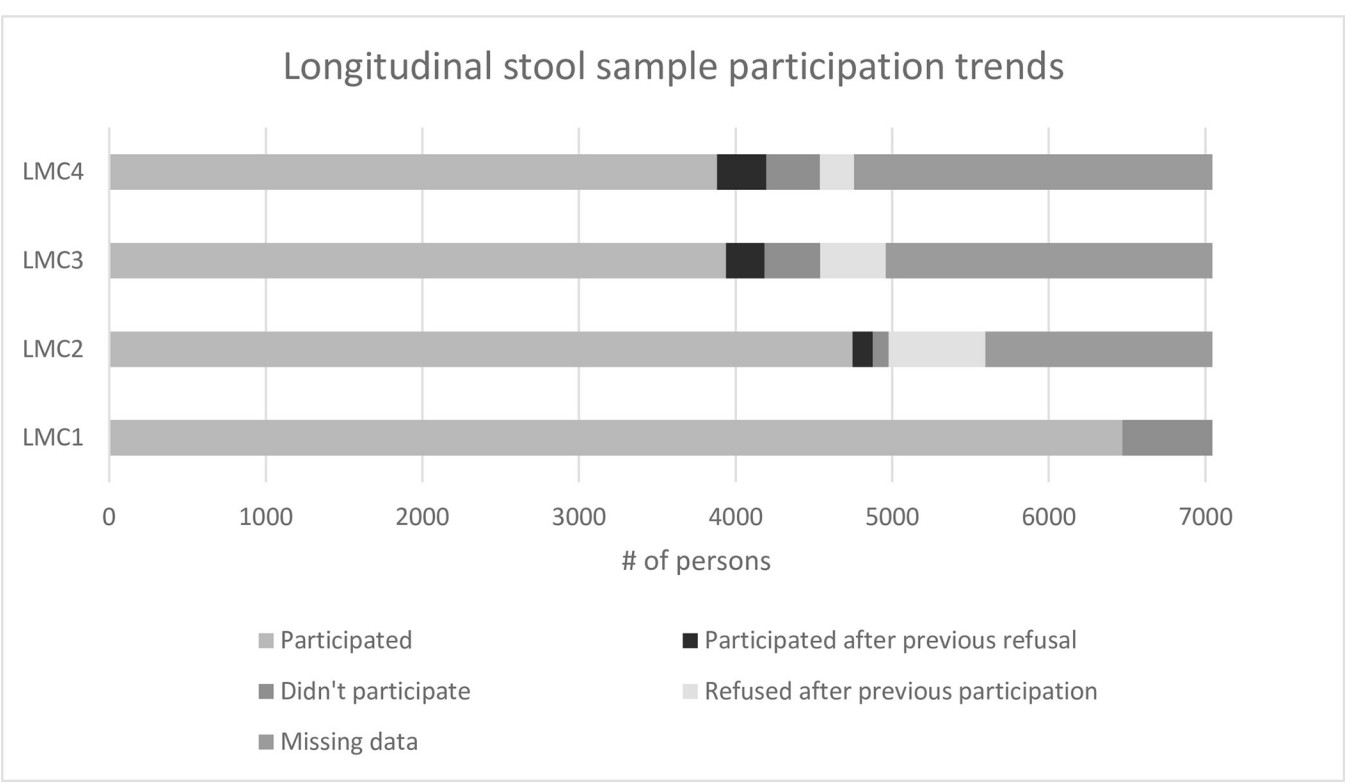

**Fig 3. Longitudinal stool sampling participation trends across longitudinal monitoring cohort (LMC) survey 1 (2018), 2 (2019), 3 (2020) and 4 (2021).**

are less willing to provide their stool sample in comparison to providing a stool sample from their child (Table 4). Adult refusals were more frequent than the refusal to provide consent for a child to provide a sample. Systematic reviews examining factors affecting participation in MDA programs for schistosomiasis and lymphatic filariasis, similarly found that adults were less likely to participate, as compared to children [29,30]. Given that adults can serve as reservoirs of infection for STH in many communities, systematic non-participation of adults in STH stool surveillance may pose a challenge to accurately estimating STH prevalence within elimination programs.

Secondly, individuals from Pédah-speaking households were less likely to refuse to provide a stool sample than other language groups. In the Beninese context, language group largely aligns with ethnicity which also drives cultural habits and norms. Furthermore, in Comé, the Pédah communities more commonly live in rural areas, in comparison to Mina and Watchi-speaking families that are concentrated in urban or peri-urban areas. Thus, language may serve as a proxy for urbanicity. This trend aligns with other quantitative findings about population density, where increased density was linked with increased refusals. Other studies exploring perspectives of MDA for both STH and schistosomiasis found that perceptions of infection risk increased participation in MDA [29,31]. In urban settings, where sanitation infrastructure tends to be more advanced and healthcare access more readily available, individuals may perceive themselves to be at lower risk of STH, and thus are more likely to refuse to participate in stool surveillance. Qualitative data from this study similarly linked experiences with STH symptoms and perception of it being a problem in the community with increased participation, supporting this hypothesis. Another explanation for why increased population density was associated with increasing refusals to provide a sample could be due to the ease with which

**Table 4. Points of convergence / divergence between qualitative and quantitative findings.**

| Qualitative Theme | Qualitative Findings | Quantitative Findings | Convergence v. Divergence |
|---|---|---|---|
| Across all clusters, community members tended to weigh community-level benefits against perceptions of individual-level risks<br>When benefits of STH surveillance for the collective are appreciated, participation increases | *"However, it is because we saw in this action a benefit for us and for our family that we became interested in it"* [Respondent 3, Men FGD, High Refusal Cluster]<br><br>*"People have agreed to provide stool samples so that the disease can be reduced."* [Respondent 1, Women FGD, Low Refusal Cluster]<br><br>*"However, it is from the unpleasant that comes the pleasant, and then as we have the love of our village, we gave the stool samples."* [Respondent 3, Men FGD, Low Refusal Cluster] | If someone has resided in their community for over 10 years, the individual was less likely to refuse to provide a sample.<br>Time in current residence [<1yr] Reference<br>Time in current residence [11-20yr] (OR:0.82, CI:0.54–1.25)<br>Time in current residence [>20yr] (OR: 0.76, CI: 0.51–1.15) | Convergence |
|  |  | If the community is an individual's main place of residence, the individual was less likely to refuse to provide a stool sample<br>(OR:0.36, CI:0.20–0.66) | Convergence |
|  |  | Villages who were in the lowest quartile of population size (<7,600 persons), had the lowest proportion of individuals refusing to provide a stool sample (5.4%) compared to villages with larger populations. | Convergence |
| Experience w/STH or perception of STH being a problem in the community increases participation | *"Especially that one of my children suffers from them. I can say that this is why I agreed to register them and provide their stool sample"* [Respondent 3, Parents FGD, High Refusal Cluster] | Individuals who refused to provide a stool sample were less likely to live in clusters with higher STH prevalence, than those who refused (3.70% and 5.36% respectively). This association between participation status and baseline cluster STH prevalence is significant (p<000.1) | Convergence |
| Rumors of misuse of stool samples were more common in high refusal clusters<br>Individuals living in communities in which rumors circulated easily had increased rates of refusal | *"Actually, the way I see my elders and my younger brothers react in the Commune since this stool sampling operation began is that in their mentality; according to them the Whites are carrying out actions to make us sick. Others go so far as to say that the Whites want to dissolve Africa, well! Things like taking their stools to use them clandestinely or to destroy them, why are they going to give the stools away! You know what I mean."* [Respondent 1, Men FGD, High Refusal Cluster] | As the population density within 1km of individuals' households increased, their likelihood of refusal also increased. The proximity of individuals to one another geographically is associated with refusal.<br>Population density [1st quartile]    Reference<br>Population density [2nd quartile]    (OR:1.40, CI:1.02–1.93)<br>Population density [3rd quartile]    (OR1.45, CI:1.01–2.08)<br>Population density [4th quartile]    (OR:1.71, CI:1.16–2.52) | Convergence |
| There is less aversion to handling children's feces, leading to increased likelihood to provide stool samples for children compared to adults | *"The worst thing is that it is not a child's stool sample that is required but an adult's, so I have refused to provide it."* [Respondent 4, Men FGD, High Refusal Cluster]<br><br>*"They say that the reason why they will not give their stool is because it is still unclean to defecate and then take their own feces and give it to investigating officers. That's why friends of mine; my own brothers and sisters refused to provide it"* [Respondent 3, Women FGD, High Refusal Cluster] | Compared to pre-school age children and school-age children, adults were more likely to refuse to participate in stool surveillance (OR:0.79, 95%CI:0.63–1.00 and OR:0.69, 95%CI:0.55–0.88, respectively) | Convergence |
| Voodoo beliefs & rumors of occult practices | *"Some were afraid and thought that their stool would be used for occult purposes. So, it was fear that motivated the refusal of those people."* [Respondent 4, Women FGD, High Refusal Cluster] | Those individuals who identified Voodoo as their household religion had a lower proportion of refusals to participate in stool surveillance (4.7%) than any other religion group (p<000.1) | Divergence |
| Changes in participation overtime | *The results no longer reached us, there is reason to be discouraged."* [Respondent 4, Parents FGD, High Refusal Cluster]<br><br>*"But, when someone goes through the experience and is imbued with the ins and outs of the operation to collect stool without inconvenience, he will be the first to approach his brother to reassure him."* [Respondent 1, Women FGD, High Refusal Cluster]<br><br>*"But there was counter-information. At the beginning, the parents accepted, some of them immediately, but later. . .when they had other information about the stool sample collection, there was a crisis of confidence, a certain atmosphere of mistrust set in."* [Respondent 4, Parents FGD, Low Refusal Cluster] | Longitudinal data demonstrate that participation varied over time. There was increased refusals overtime with a large drop-off in participation after the first round of stool surveillance. Additionally, some individuals who initially refused to participate later chose to participate and vice versa. (Fig 3) | Convergence |

rumors about misuse of stool samples and misinformation about the potential dangers of providing a sample can circulate in communities with high population density. These differences in participation are particularly concerning as rural-to-urban migration increases; if urban populations are more likely to refuse consent for STH surveillance, programs will need to address this.

An important driver of participation in stool-based surveillance identified by this study is recognition of benefits for the collective community. FGD respondents who provided a stool sample reported prioritizing community-level benefits (e.g. that surveillance will help inform treatment programs and reduce STH prevalence in the community) and that these benefits outweighed individual aversion to handling feces and other perceived individual-level risks. In contrast, refusal to provide a sample was largely driven by mistrust of the sample's purpose and fear about misuse or physical harm to the individual. These results parallel findings from other settings that tie decisions to participate in STH programs, willingness to provide a stool sample for diagnostic purposes, and consent to NTD surveillance activities to an understanding of the importance of the research, fears of how the samples might be misused, and perceived benefits [15–17,19]. Similarly, studies examining participation in MDA have found that trust in the program, rumors about harmful consequences, and perceptions of benefits drive acceptability of treatment [29–32]. Of note, findings from this study align with evidence from high-income settings where patients faced barriers to providing a stool sample in a primary healthcare setting due to a lack of information and concerns about privacy [17]. That individuals are acting on their belief of the intervention's benefit for the community is linked with quantitative findings that suggest an influence of community cohesion/identity on participation. Individuals who resided in the community at least eleven years were less likely to refuse to provide a sample, suggesting the importance of community identity for participation in public health surveillance. Similarly, individuals who did not reside in the community for the majority of days in the six months preceding the survey were substantially more likely to refuse to participate. Interventions targeting improved community building may be important for increasing participation in public health surveillance activities that primarily afford community benefit. Table 4 presents a full matrix of points of convergence (and divergence) between the qualitative and quantitative findings.

Several concrete recommendations for STH surveillance programs were generated from this work. If STH surveillance is intensified to achieve and document progress towards elimination as a public health problem and/or transmission interruption, programs will need to understand who is not participating in each setting. Programs will need to disaggregate refusal data based on key demographics to understand if certain populations are refusing to provide a sample at higher rates. In addition to understanding who is refusing to participate in STH surveillance, programs will need to better understand why those individuals or groups are hesitant. Investing in learning about community perspectives, priorities, and beliefs about public health surveillance will be key to ensuring the program's success. Ideally such research should be conducted in advance of large surveillance campaigns to increase acceptability. These findings also suggest programs should focus on addressing misinformation and demonstrating the benefits of stool surveillance for a community, even where perceived risk of STH is low. This will have implications for community engagement strategies, training for sample collectors, and communication messages. Programs thus have an opportunity to reimagine how STH surveillance is conducted to address community concerns and ensure groups are equally represented in surveillance data.

## Limitations

This study has several limitations. First, a high proportion (70%) of individuals sampled for the FGD were unreachable, unwilling, or unable to participate. This sampling frame may skew

findings, especially if individuals not included in the FGD were also more likely to refuse to participate in the study. Secondly, due to limitations in time and budget, this study was unable to conduct a member checking exercise, wherein main findings are shared with FGD participants for feedback and confirmation. However, the research team in Benin drew from their field experience and cultural knowledge to validate the themes emerging from the qualitative analysis to address this limitation. FGDs may also inadvertently introduce social desirability bias due to the group format of data collection. Quantitative data collection tools were developed prior to the qualitative analysis; thus, there may be some drivers of participation identified in the qualitative data that were not explored quantitatively, such as previous experience providing a stool sample. In the quantitative analysis, the proxy measures for community solidarity may be imperfect measures. Further studies are warranted to identify strong indicators of community solidarity. Additionally, while the choices made for this analysis are appropriate for exploratory research questions, it should be noted that using stepwise regression to build the quantitative model and interpretating results from a multivariate analysis have their limitations [33,34]. This analysis drew on data only from the Comé district in Benin. Some findings, particularly those pertaining to local cultural customs or beliefs, may not be generalizable to other settings.

## Conclusion

This study found that certain groups of individuals are more likely to refuse to participate in STH surveillance activities than others. Our findings suggest that adults, persons who do not reside full time in the community, newly arrived members to the community, individuals who do not perceive themselves or their communities to be at risk for STH, and individuals exposed to convincing misinformation are more likely to refuse to provide a stool sample. STH surveillance must achieve high levels of representative participation; otherwise, programs could misidentify priority areas for implementation or bias verification data. This is of particular concern if groups refusing to participate in surveillance also refuse to participate in deworming campaigns.

## Supporting information

**S1 Fig. Conceptual model for participation in STH surveillance.**
(TIF)

**S1 Appendix. FGD question guide.**
(DOCX)

**S2 Appendix. Codebook.**
(DOCX)

**S3 Appendix. AIC selection steps.**
(DOCX)

## Acknowledgments

The authors wish to thank all of the study participants, communities, community leaders, national NTD program staff, and local, regional, and national partners (Programme National de lutte contre les Maladies Transmissibles du Ministère de la Santé du Bénin) who have participated in or supported the implementation of the DeWorm3 study. We thank and acknowledge the entire DeWorm3 team in Benin for their work in data collection and study implementation. We also thank and recognize Sean Galagan, Kristjana Asbjörnsdóttir, Emily

Pearman, and Ken Tapia for providing consultations on data and quantitative data analysis. Mitsuko Hasegawa supported the development of the initial project design and qualitative data procedures. Additional consultation was sought from the Center for Studies in Demography & Ecology at the University of Washington (Eunice Kennedy Shriver National Institute of Child Health and Human Development research infrastructure grant, P2C HD042828).

## Author Contributions

**Conceptualization:** Emma Murphy, Moudachirou Ibikounlé, Judd L. Walson, Arianna Rubin Means.

**Formal analysis:** Emma Murphy, Innocent Comlanvi Togbevi.

**Investigation:** Innocent Comlanvi Togbevi, Moudachirou Ibikounlé, Euripide FGA Avokpaho.

**Methodology:** Emma Murphy, Innocent Comlanvi Togbevi, Arianna Rubin Means.

**Supervision:** Moudachirou Ibikounlé, Judd L. Walson, Arianna Rubin Means.

**Writing – original draft:** Emma Murphy.

**Writing – review & editing:** Emma Murphy, Innocent Comlanvi Togbevi, Moudachirou Ibikounlé, Euripide FGA Avokpaho, Judd L. Walson, Arianna Rubin Means.

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
