## [Decision Letter · Decision Letter 0]

9 Sep 2022

Dear Dr. Means,

Thank you very much for submitting your manuscript "Soil-transmitted helminth surveillance in Benin: a mixed-methods analysis of factors influencing non-participation in longitudinal STH surveillance activities" for consideration at PLOS Neglected Tropical Diseases. As with all papers reviewed by the journal, your manuscript was reviewed by members of the editorial board and by several independent reviewers. The reviewers appreciated the attention to an important topic. Based on the reviews, we are likely to accept this manuscript for publication, providing that you modify the manuscript according to the review recommendations. 

Sincerely,

Samuel Wanji

Academic Editor

Francesca Tamarozzi

Section Editor

Reviewer's Responses to Questions

**Key Review Criteria Required for Acceptance?**

**Methods**

-Are the objectives of the study clearly articulated with a clear testable hypothesis stated?

-Is the study design appropriate to address the stated objectives?

-Is the population clearly described and appropriate for the hypothesis being tested?

-Is the sample size sufficient to ensure adequate power to address the hypothesis being tested?

-Were correct statistical analysis used to support conclusions?

-Are there concerns about ethical or regulatory requirements being met?

Reviewer #1: The authors may consider stating quantitative and qualitative research questions

Were survey weights, strata considered in the analysis?

A directed acyclic graph demonstrating causal relationships between variables could add clarity what independent variables were considered for the inclusion.

Reviewer #2: a) Line 132: rather than “QUAL � QUANT” this could be explained as qualitative followed by quantitative research 

b) Paragraph beginning line 157: It is explained later in the quantitative section, but it would be useful to clarify here that it was possible to agree to participate in the LMC but then refuse to give a stool sample (i.e. there were two types of refusal: at the time of recruitment and at the time of giving the sample) 

c) Line 159: please clarify how the purposive sampling was done- e.g. was it done by visiting households? Who carried out the sampling? How were the potential participants identified to take part? 

d) Table 1- please add how many participants gave a sample/ didn’t give a sample in each FGD, so as to better understand the composition of the groups

e) Paragraph starting line 173: the authors should consider including a reflexivity/ positionality statement in this section, to explain how they are situated in relation to the research participants and analytical processes

f) Line 194: please clarify the term “case memo”, does this mean a summary of themes? 

g) Line 217: please give details of the methods used to check for multicollinearity 

h) Line 224: this is the first mention of ‘statistical significance’, but this comment applies to other parts of the manuscript where this concept is invoked. I recommend that the authors take note of the American Statistical Association’s guide to t p-values (https://www.ncbi.nlm.nih.gov/pmc/articles/PMC5187603/). The advice in this reference may be used, but in general I recommend that the authors: 

i. Avoid using the arbitrary cut-off of 0.05 to determine ‘significance’ 

ii. Avoid concluding that is a variable has a p-value of less than 0.05 it is therefore ‘important’ (and if it is above 0.05 it is therefore ‘unimportant’) 

iii. Avoid bolding or highlighting p-values below 0.05

iv. Give the p-value along with a 95% CI and comment on the magnitude of the observed effect 

i) Line 225: it is good practice to cite some of the R packages that were used, and I recommend that the authors do so. Please see this guidance for more detail: https://ropensci.org/blog/2021/11/16/how-to-cite-r-and-r-packages/

j) There are a few things it would be beneficial to mention in the methods section: 

i. Who moderated the FGDs and how were they trained in advance? 

ii. How was missing data dealt with? 

iii. How was socio-economic status calculated? 

iv. How was population density measured? Was it using data from the wider trial, or was it from national statistics?

v. Mention of ethical approval

Reviewer #3: You may want to use “factors associated with” to replace “correlates”

**Results**

-Does the analysis presented match the analysis plan?

-Are the results clearly and completely presented?

-Are the figures (Tables, Images) of sufficient quality for clarity?

Reviewer #1: (No Response)

Reviewer #2: a) Line 293: this title doesn’t reflect the disconnection from the hospital system that participants reported- recommend that this is updated to reflect this element of the following section

b) Quote from line 308: this doesn’t really speak to the themes in the preceding paragraph, authors should consider removing or replacing the quote

c) Quote from line 404: this is about inclusion of local people in activities, which is not mentioned in the above paragraph, I suggest that this is rectified 

d) Paragraph from line 420: rather than listing which variables had a p-value less than 0.05, a more descriptive approach should be taken to characterising the sample

e) Table 3- please add a reference category for living in community during last 6 months 

f) Table 3- it is not recommended to bold values which are below 0.05, as it dichotomises the interpretation of p-values based on an arbitrary cut-off point

g) Figure 3: the terms LMC1-4 should be explained below the figure

Reviewer #3: Please provide the method and results of table 4 to the methods and results section.

Please provide detailed information for the “First, a high rate (70%) of sampled individuals were 531 unreachable, unwilling, or unable to participate in FGDs.” in the results section

**Conclusions**

-Are the conclusions supported by the data presented?

-Are the limitations of analysis clearly described?

-Do the authors discuss how these data can be helpful to advance our understanding of the topic under study?

-Is public health relevance addressed?

Reviewer #1: Concerning the study limitations:

Given the high non-response rate in the study, have the authors considered other ways to check whether non-respondents and respondents were similar or different?

If the quantitative data collection tools were developed before the qualitative study part was conducted, then the study design is not sequential anymore?

Reviewer #2: 1. Discussion

a) Paragraph from 467- this background information is not needed at the start of the Discussion, I recommend that the discussion begins with line 473 (“This paper presents…”) 

b) If possible, it would be useful to see a discussion of the non-participation percentage of 8.1%. Are there any other studies to compare to? Is this high or low, comparatively, for STH surveillance activities? 

c) Line 522: some further implications for practice from this research should be given here. What should future groups undertaking surveillance activities do to ensure higher participation? Based on the authors’ results, it would be beneficial to draw out the practical recommendations in more detail (especially as the results are so rich and actionable) 

d) Table 4: This table is useful, but should be moved either to the end of the Results section or to the Supplementary Materials. As it stands, it does not fit into the narrative of the Discussion section 

2. Limitations

a) Line 534: Please clarify what is meant by a “member checking exercise” 

b) Line 534: “Validation of the themes emerging from the qualitative analysis was conducted by the research team in Benin”- this is a strength of the work, it should be removed from this section

c) Some further limitations should be noted: 

i. The use of FGDs: although the use of FGDs was an appropriate choice for this study, it should be acknowledge that they may introduce social desirability biases (for example, members of the group may defer to those of higher status and not voice dissenting opinions)

ii. It should be acknowledged that the proxy measures for community solidarity used in the quantitative element (while still a valid choice) are imperfect

iii. While stepwise regression was an appropriate method for an exploratory research question with little previous theory to structure alternative modelling strategies, its deficiencies should be acknowledged. See, for example, the relevant sections in Regression Modelling Strategies (2006) by Frank E Harrell 

iv. Similarly, while it was appropriate in this case as the study was exploratory, there are drawbacks to presenting multiple adjusted effect estimates from a single model in a table, which should be acknowledged. See this paper on the Table 2 Fallacy for further information: https://academic.oup.com/aje/article/177/4/292/147738

3. Conclusions 

a) I suggest that the authors reduce the conclusion to one paragraph, and remove some of the additional detail (for example, lines 550-553) 

b) The paragraph starting from line 554 contains recommendations for practice- these should be introduced in the Discussion section, and summarised in the Conclusion

Reviewer #3: (No Response)

**Editorial and Data Presentation Modifications?**

Reviewer #1: (No Response)

Reviewer #2: a) Recommend that p-values are not bolded in the Tables 

b) Figure 3: the terms LMC1-4 should be explained below the figure

Reviewer #3: (No Response)

**Summary and General Comments**

Reviewer #1: Overall, this is a well-written manuscript and has a scientific merit. Used methods are sufficiently described and are aligned with the study aim and results.

Reviewer #2: This is an article which aimed to understand barriers to participation in stool-based surveillance for soil-transmitted helminths in Benin. Using a mixed methods approach, it found that the perception of community benefit vs individual risk, the impact of rumours, interactions with field agents, norms around handling faeces, age and community affinity influenced participation. 

There is much to commend in this well-written article. The study asks a specific and important question which has clear implications for public health practice. The choice of a mixed-methods approach provided illuminating results, and the theoretical underpinning of the work has clearly been well thought through. The comments made are intended to add further detail and clarification. 

1. General comments 

a) The authors sometimes use Benin and at other times Bénin, please align on one spelling 

2. Title

a) To avoid use of an abbreviation in the title, suggested update to: “A mixed methods analysis of factors influencing non-participation in longitudinal soil-transmitted helminth surveillance in Benin”

3. Abstract 

a) Line 37: “non-participation” rather than “non-compliance” may be more appropriate here to ensure terminology is consistent throughout 

b) Lines 38-39: as the article is about non-participation, it may be better to phrase this as an investigation of drivers of non-participation or barriers to participation, rather than drivers of participation (this occurs elsewhere in the abstract) 

c) Line 40: suggest to simplify to “exploratory mixed-methods study” 

d) Line 44: should clarify where the data for the mixed-effects logistic regression came from

e) Line 49: on reading the rest of the article, it seems like adults were more willing to provide stool samples from children than from themselves, rather than children were more willing to provide samples in comparison with adults. Would be useful to clarify this point (which comes up elsewhere in the manuscript), which may include a Discussion of consent/ assent procedures used in the DeWorm3 project 

4. Introduction 

a) Line 93: suggest to clarify what these existing STH surveillance protocols are

b) Line 95: by “delineating areas with low baseline transmission”, do the authors mean identifying areas that do not need to be targeted by enhancing surveillance and treatment methods? This phrase may need some clarification

c) Lines 199-120: is the claim that specific factors are unknown accurate? The Kenya study referenced above in the same paragraph gives some indication of factors influencing the decision to give a stool sample for STH surveillance. Suggestions that this is reworded to say we don’t have a full understanding of the issue, rather than that this is completely unknown 

d) Line 121-122: are there any studies to support this claim? Fine if not, as it is a reasonable assumption, but would be good to have some view of how wide-spread this problem is 

5. Supplementary Materials 

a) The cluster refusal rates and longitudinal trends figures are included in the main body of the manuscript and so don’t need to be included in the Supplementary Materials as well

Reviewer #3: (No Response)

PLOS authors have the option to publish the peer review history of their article (what does this mean?). If published, this will include your full peer review and any attached files.

Reviewer #1: Yes: Alpamys Issanov

Reviewer #2: No

Reviewer #3: No

Figure Files:

Data Requirements:

Reproducibility:

References

---

## [Decision Letter · Decision Letter 1]

25 Nov 2022

Dear Dr. Means,

We are pleased to inform you that your manuscript 'Soil-transmitted helminth surveillance in Benin: a mixed-methods analysis of factors influencing non-participation in longitudinal surveillance activities' has been provisionally accepted for publication in PLOS Neglected Tropical Diseases.

Best regards,

Samuel Wanji

Academic Editor

Francesca Tamarozzi

Section Editor

Reviewer's Responses to Questions

**Key Review Criteria Required for Acceptance?**

**Methods**

-Are the objectives of the study clearly articulated with a clear testable hypothesis stated?

-Is the study design appropriate to address the stated objectives?

-Is the population clearly described and appropriate for the hypothesis being tested?

-Is the sample size sufficient to ensure adequate power to address the hypothesis being tested?

-Were correct statistical analysis used to support conclusions?

-Are there concerns about ethical or regulatory requirements being met?

Reviewer #1: (No Response)

Reviewer #2: (No Response)

**Results**

-Does the analysis presented match the analysis plan?

-Are the results clearly and completely presented?

-Are the figures (Tables, Images) of sufficient quality for clarity?

Reviewer #1: (No Response)

Reviewer #2: (No Response)

**Conclusions**

-Are the conclusions supported by the data presented?

-Are the limitations of analysis clearly described?

-Do the authors discuss how these data can be helpful to advance our understanding of the topic under study?

-Is public health relevance addressed?

Reviewer #1: (No Response)

Reviewer #2: (No Response)

**Editorial and Data Presentation Modifications?**

Reviewer #1: (No Response)

Reviewer #2: (No Response)

**Summary and General Comments**

Reviewer #1: The authors addressed all my comments. I minor, not important comment - will leave them to decide

I think the first paragraph in the Methods section could be moved to the introduction as it explains the study aims and the proposed methodology.

Reviewer #2: Thank you for the opportunity to review the revised manuscript. The authors have addressed all of my previous comments thoroughly and thoughtfully, so I recommend that the article is accepted.

PLOS authors have the option to publish the peer review history of their article (what does this mean?). If published, this will include your full peer review and any attached files.

Reviewer #1: No

Reviewer #2: No

---

## [Editor Report · Acceptance letter]

21 Dec 2022

Dear Ms. Murphy,

We are delighted to inform you that your manuscript, "Soil-transmitted helminth surveillance in Benin: a mixed-methods analysis of factors influencing non-participation in longitudinal surveillance activities," has been formally accepted for publication in PLOS Neglected Tropical Diseases.

Best regards,

Shaden Kamhawi

co-Editor-in-Chief

Paul Brindley

co-Editor-in-Chief
